# MODELING RELATIONAL TIME SERIES USING GAUSSIAN EMBEDDINGS

**Ludovic Dos Santos,**\***Ludovic Denoyer, Benjamin Piwowarski & Patrick Gallinari**
Sorbonne Universities, UPMC Univ Paris 06, CNRS, LIP6 UMR 7606
4 place Jussieu 75005 Paris, France
`firstname.lastname@lip6.fr`

**Ali Ziat**\*
Sorbonne Universities, UPMC Univ Paris 06, CNRS, LIP6 UMR 7606
Institut VEDECOM, 77 rue des chantiers, 78000, Versailles
`ali.ziat@vedecom.fr`

## ABSTRACT

We address the problem of modeling multiple simultaneous time series where the observations are correlated not only inside each series, but among the different series. This problem happens in many domains such as ecology, meteorology, etc. We propose a new dynamical state space model, based on representation learning, for modeling the evolution of such series. The joint relational and temporal dynamics of the series are modeled as Gaussian distributions in a latent space. A decoder maps the latent representations to the observations. The two components (dynamic model and decoder) are jointly trained. Using stochastic representations allows us to model the uncertainty inherent to observations and to predict unobserved values together with a confidence in the prediction.

## 1 INTRODUCTION

Relational time series, i.e. multiple time series where the observations are correlated both inside each series and between series occur in many domains such as ecology, medicine, biology, earth observation by satellite imagery or local measurements, multimedia or even social data analysis. The correlations between the different observed series can come from a proximity (e.g. earth observation or epidemic diffusion) or from a similarity of behavior (e.g. user traces in social data). In the statistical literature, the modeling of relational time series has been the topic of a dedicated field: spatio-temporal statistics (Cressie & Wikle (2011); Wikle & Hooten (2010)). Different methodologies have been developed for handling a large variety of spatio-temporal phenomena, with an emphasis on the analysis of natural observations like weather prediction, ecology or remote sensing. In the machine learning domain, there exists a vast literature dedicated to sequence or time series prediction. Recently, deep recurrent neural networks have witnessed notable successes in different sequence and time series modeling tasks leading to an increasing number of publications, e.g. (Barbounis et al. (2006); Hsieh et al. (2011); Cao et al. (2012); Hermans & Schrauwen (2013)). Despite a large number of recent developments, the modeling and analysis of relational time series has only attracted a few attention in the field of representation learning. In addition, most of the models are deterministic in the sense that they are trained to learn a fixed mapping for modeling the dynamics of the series.

We propose a new state space model for relational time series able to model the uncertainty at the observation and at the modeling levels. The principle of this approach is to associate each point of a time series to a Gaussian distribution in a latent space, the distribution over the observed values being directly computed from these latent distributions. The model has two main components. One is responsible for the dynamics in the latent space. This component is thus modeling the evolution of the Gaussian distribution considering both the temporal intra-series and the relational inter-series

---

\*Both authors contributed equally to this work

dependencies. A second component acts as a decoder and maps the latent representations associated with each series to the corresponding observations in the output space.

The contributions of the paper are thus: (i) a new dynamical model for relational time series inspired by representation learning; (ii) a stochastic component for modeling the uncertainties at the observation and dynamic levels

The paper is organized as follows. In Section 2 we introduce some related work on forecasting in time series, representation learning for time series, and recent deep learning works focusing on modeling uncertainty. The model is presented in Section 3 together with four different variants. Section 4 presents experimental results on four datasets, and section 5 concludes this work and gives some perspectives.

## 2    RELATED WORK

The classical topic of time series modeling and forecasting has given rise to an extensive literature. In statistics, classical linear models include many variations around auto-regressive and moving average models (De Gooijer & Hyndman (2006)). In machine learning, non linear extensions of these models based on neural networks have been proposed as early as the 90s, opening the way to many other non linear models including kernel methods (Muller et al. (99)).

Relational time series have mainly been studied in the field of spatio-temporal statistics (Cressie & Wikle (2011); Wikle & Hooten (2010)). The traditional method first relied on a descriptive approach using the first and second-order moments of the process for modeling the spatio-temporal dependencies. More recently, dynamical state models, where the current state is conditioned on the past have been explored (Wikle (2015)). These models have been considered both for continuous/discrete space and time components. However, the most common way is to consider discrete time, leading to the modeling of time series of spatial processes as we do here. When space is discrete, the model comes down to a general vectorial autoregressive formulation. These models face a curse of dimensionality in the case of a large number of sources. Different strategies have been adopted to solve this problem such as embedding the spatio-temporal process in a low-dimensional manifold or parameter reduction (Wikle (2015)), leading to model families quite similar to the ones used in machine learning for modeling dynamical phenomena. Also, for complex underlying processes, observations only provide an incomplete description of the process dynamics so that modeling uncertainty at the data and model levels is an important topic.

In the last 10 years, there has been a growing interest in learning latent representations for example through neural networks and deep learning. Dynamical state space models such as recurrent neural networks (RNN), which have been used for time series forecasting in different contexts since the early nineties (Connor et al. (1994)), have recently witnessed important successes in different areas for general sequence modeling problems, leading to breakthroughs in domains like speech (Graves et al. (2013)), language generation (Sutskever et al. (2011)), translation (Cho et al. (2014)), and many others. Among this family, the model closest to ours is the dynamic factor graph model of (Mirowski & LeCun (2009)) designed for multiple series modeling for the tasks of forecasting and imputation. However this model does not consider relational dependencies which is the focus of our approach.

Most of the above models make use of pointwise representations and do not model explicitly the uncertainties present in the process and/or in the observations. Recently, in the learning representation community, there has been a growing interest in using distributions as latent representations instead of points. (Vilnis & McCallum (2015); He et al. (2015); Dos Santos et al. (2016)) all make use of Gaussian distributions for representing different items like words (Vilnis & McCallum (2015)), nodes in knowledge graphs (He et al. (2015)) or nodes in graphs for transductive classification (Dos Santos et al. (2016)). Note that Gaussian processes have also been used for time series prediction, but they have mainly been considered for univariate time series prediction (Hachino & Kadirkamanathan (2011); Brahim-Belhouari & Bermak (2004)) and they do not use a state space formulation.

Recent techniques in variational inference (Kingma & Welling (2014); Rezende et al. (2014)) deal with uncertainty by modeling distributions in the observation space, mapping random variables within a latent space to observations with a deep neural network. Extension of the variational in-

ference method to time series has been proposed (Fraccaro et al. (2016); Krishnan et al. (2015)) but contrarily to those works, we take into account relationships (both temporal and relational). Furthermore, in our model, we work directly with random variables to predict observations from time series. This gives us direct access to the output distribution with no need to sample or work with intractable distributions.

Our model is built on top of the model in (Ziat et al. (2016)) which proposes a deterministic dynamical process model but does not consider any explicit modeling of uncertainty. In this paper, we propose a model that uses Gaussian embeddings, and extend the dynamics and loss functions of the model in (Ziat et al. (2016)).

## 3 FORECASTING OF RELATIONAL TIME SERIES

### 3.1 NOTATIONS AND TASKS

Let us consider a set of $n$ temporal sequences[1] $\mathbf{x_1}, .., \mathbf{x_n}$ such that $x_i^{(t)} \in \mathbb{R}$ is the value of the $i$th sequence at time $t$ defined by $\mathbf{x_i} = (x_i^{(1)}, .., x_i^{(T)})$, $T$ being the number of observed time steps. For simplification, we consider that all the series have the same length, but this is not restrictive.

We model the dependencies between the different series through a graph, the different series sources being the graph vertices and the links modeling explicit dependencies between the sources. These links can reflect a spatial proximity between the sources of the series, a similarity of behavior between users or any other predefined relation. These explicit relations will be modeled in the latent space. Our hypothesis is that they will constrain the representation of linked sources to be similar one to another in the latent space, this similarity being controlled by the strength of the link between the two time series, denoted $e_{i,j}$. We assume that the graph structure is static in time and is provided as a prior information. The model can be extended to learn these static dependencies but this is not considered here.

Let us denote $\tau$ the size of the prediction horizon. The **forecasting** problem considered here is to compute for all series $i$ the values $x_i^{(T+k)}$ for all $k$ in $[1; \tau]$. Note that the model can be straightforwardly extended to the imputation problem that aims at predicting missing values.

### 3.2 INFORMAL DESCRIPTION

The proposed model is a dynamic state space model: the dynamics is modeled in a continuous latent state space and the observations are generated from states in this latent space. State space models have already been considered for multiple time series (e.g. Mirowski & LeCun (2009)) and for spatio-temporal processes (e.g. Wikle & Hooten (2010)).

Both the observations and the dynamics are subject to uncertainties. Usually, the observations correspond to a partial view of the underlying generating process and the dynamics being hidden is not directly accessible and should be modeled as a stochastic process.

To handle this uncertainty, we propose a model, namely Relational Dynamic model with Gaussian representations (**RDG**), that represents latent factors as **distributions** in a latent space and learns the series dynamics in this latent space. The distributions themselves are estimated using observations like for any other representation learning model. Besides being more adapted to handling the noise inherent to the process and to the observations, the model can be used to predict the posterior distribution of the variables associated to the series and in particular the confidence or variance associated to the predictions.

The model is an extension of the deterministic model of (Ziat et al. (2016)) and has two main components: (i) **Decoding component:** we consider that each series corresponds to a particular *trajectory* in an unknown latent space. Each series $x_i^{(1)}, ...., x_i^{(T)}$ is thus associated to a series of random variables in $\mathbb{R}^d$ denoted $Z_i^{(1)}, ...., Z_i^{(T)}$, $Z_i^{(t)}$ being the latent factor explaining the observed value of the series $i$ at time $t$ and $d$ the size of the latent space. We model each $Z_i^{(t)}$ as a multivariate

---

[1]For simplicity, we consider univariate time series, but the model can be trivially extended to multivariate time series.

normal variable $\mathcal{N}(\mu_i^{(t)}, \Sigma_i^{(t)})$. The observation can be computed from this latent distribution by using a *decoding function* mapping $Z_i^{(t)}$ to $X_i^{(t)} = f(Z_i^{(t)})$. (ii) **Dynamic component:** The second component models the series dynamics in the latent space. We suppose that dynamics can be captured for all series through a function $h$ that maps the latent random variable $Z_i^{(t)}$ to the next latent variable $Z_i^{(t+1)} = h(Z_i^{(t)})$. The function $h$ is thus modeling the time dynamics. In addition, constraints are introduced to reflect prior knowledge about the relational dependency structure of the series. For any couple of series $i$ and $j$ with a known dependency, i.e. such that $e_{i,j} > 0$ we add a corresponding constraint on $Z_i^{(t)}$ and $Z_j^{(t)}$ as explained in Section 3.3.3.

In the following, we explain how the distributions corresponding to the random variables $Z_i^{(t)}$ are learned, jointly to the functions $f$ (decoder component) and $h$ (dynamic component).

## 3.3 MODEL DEFINITION

We suppose that the random variables $Z_i^{(t)}$ follow a Gaussian distribution. Let us denote $Z_i^{(t)} \sim \mathcal{N}(\mu_i^{(t)}, \Sigma_i^{(t)})$ a distribution where $\mu_i^{(t)}$ and $\Sigma_i^{(t)}$ have to be estimated from known observations. For simplicity, we consider in the following that $\Sigma_i^{(t)}$ is a diagonal matrix, with $\sigma_{i,j}^{(t)}$ denoting the $j$th value of the diagonal of $\Sigma_i^{(t)}$.

We define a global loss function $\mathcal{L}(\mu, \Sigma, f, h)$ where $\mu$ and $\Sigma$ are the means and covariance matrices for all the series and for all the time steps between $1$ and $T$. The loss is a sum of three terms: (i) a decoding loss $\Delta_{\text{De}}$, (ii) a dynamical loss $\Delta_{\text{Dy}}$ and (iii) a structural loss $\Delta_{\text{R}}$:

$$\mathcal{L}(\mu, \Sigma, f, h) = \sum_{i=1}^{n} \sum_{t=1}^{T} \Delta_{\text{De}}(f(Z_i^{(t)}), x_i^{(t)}) + \lambda_{\text{Dy}} \sum_{i=1}^{n} \sum_{t=1}^{T-1} \Delta_{\text{Dy}}(Z_i^{(t+1)}, h(Z_i^{(t)}))$$
$$+ \lambda_{\text{R}} \sum_{j=1}^{n} \sum_{t=1}^{T} e_{i,j} \Delta_{\text{R}}(Z_i^{(t)}, Z_j^{(t)}) \quad (1)$$

where $\lambda_{\text{Dy}}$ and $\lambda_{\text{R}}$ are hyperparameters weighting the importance of the different elements in the loss function. The first term corresponds to the *decoding component*, and forces both $f$ and the learned distributions of variables $Z$ to "explain" the observations, the second term, the *dynamic component*, encourages $h$ to model the time dynamics in the latent space, while the third term captures the relations between the pairs of series. In the following, we use for $f$ a **linear function** and $h$ will be either a linear or non-linear function (see Section 3.3.2).

**Learning:** Learning the model is performed through the minimization of the loss function $\mathcal{L}(\mu, \Sigma, f, h)$ with respect to $\mu$, $\Sigma$, $f$ and $h$. To simplify the notations, the parameters of $f$ and $h$ are not made explicit in the notations – $f$ and $h$ are supposed to be differentiable. At the end of the learning process, all the latent distributions for each of the time steps are known for the training data, as well as the decoding function $f$ and the dynamical one $h$. We used *ADAM* (Kingma & Ba (2015)) as a stochastic gradient descent technique. This optimization can be easily made on a large scale dataset, and/or by using GPUs.

### 3.3.1 FROM LATENT SPACE TO OBSERVATIONS

The mapping onto the latent space is learned so that the values $x_i^{(t)}$ of each series can be predicted from their respective Gaussian embedding $Z_i^{(t)}$ through the $f$ function. We define below two alternative decoding loss functions $\Delta_{\text{De}}$, used in the experiments for measuring the error between the predicted distribution $f(Z_i^{(t)})$ and the observation $x_i^{(t)}$. Other losses could be used with the same model.

The **first loss** measures the difference between the expected value of $f$ and the observation using a mean-square error:

$$\Delta_{\text{De}_1}(f(Z_i^{(t)}), x_i^{(t)}) \stackrel{\text{def}}{=} \left( E\left[ f(Z_i^{(t)}) \right] - x_i^{(t)} \right)^2 \quad (2)$$

When considering a linear decoding function such as $f(\cdot) = <\theta, \cdot>$, $\theta$ being the set of parameters of $f$, $\Delta_{\text{De}_1}$ can be rewritten as as:

$$\Delta_{\text{De}_1}(f(Z_i^{(t)}), x_i^{(t)}) = (<\theta, \mu_i^{(t)}> -x_i^{(t)})^2 \tag{3}$$

The **second loss** aims at measuring the distance between the random variable modeling the predicted observations and the observations. This is the expectation of the mean squared error between the predictions and the observations:

$$\Delta_{\text{De}_2}(f(Z_i^{(t)}), x_i^{(t)}) \overset{\text{def}}{=} E\left[(f(Z_i^{(t)}) - x_i^{(t)})^2\right] \tag{4}$$

When $f$ is a linear function, this loss can be written as:

$$\Delta_{\text{De}_2}(f(Z_i^{(t)}), x_i^{(t)}) = \sum_{k=1}^{d} \theta_k^2 \sigma_{i,k}^{(t)} + \left(<\theta, \mu_i^{(t)}> -x_i^{(t)}\right)^2 \tag{5}$$

Minimizing $\Delta_{\text{De}_1}$ only updates the mean of the distributions, whereas minimizing $\Delta_{\text{De}_2}$ updates both the mean and the variance. More specifically, an observed value with $\Delta_{\text{De}_2}$ will pull the variances $\sigma_i^{(t)}$ down. This is an interesting property since observing values should reduce the variance of the representation. Moreover, this effect will be higher for the dimensions of the latent space where the value of $\theta$ is higher. This is sensible since variance is reduced for the dimensions that are important for the prediction.

### 3.3.2 MODELING DYNAMICS

The loss function $\Delta_{\text{Dy}}$ aims at finding values $Z_i^{(\cdot)}$ and a dynamic model $h$, that will be used to predict the representation of the next state of time series $i$, $Z_i^{(t+1)}$. The function $h$ maps a distribution $\mathcal{N}(\mu_i^{(t)}, \Sigma_i^{(t)})$ to $\mathcal{N}(\mu_i^{(t+1)}, \Sigma_i^{(t+1)})$. Based on (Vilnis & McCallum (2015); Dos Santos et al. (2016)), we use a Kullback-Leibler divergence (noted $D_{KL}(\cdot||\cdot)$) to compare the distribution at $(t+1)$ to the distribution predicted by $h$.

We propose in the following two alternative functions for $h$. For the first one, we consider that the latent representation at time $(t+1)$ is a linear transformation of the latent distribution at time $t$. The transformed variable is also a Gaussian and its parameters can be easily computed. In this case, $h$ is a linear function from $\mathbb{R}^d$ to $\mathbb{R}^d$ which is represented by a matrix $\gamma \in \mathcal{M}_{d,d}(\mathbb{R})$:

$$\Delta_{\text{Dy}_1}(Z_i^{(t+1)}, h(Z_i^{(t)})) \overset{\text{def}}{=} D_{KL}(Z_i^{(t+1)}||\gamma Z_i^{(t)}) = D_{KL}(Z_i^{(t+1)}||\mathcal{N}(\gamma\mu_i^{(t)}, \gamma\Sigma_i^{(t)}\gamma^T)) \tag{6}$$

Linear transformations of random vectors might be too restrictive to model complex processes. As an alternative transformation, we used two non linear multilayer perceptrons (MLP), one $h^m$ for predicting the means and one for $h^c$ for predicting the variance: the next mean is given by $\mu_i^{(t+1)} = h^m(\mu_i^{(t)}, \Sigma_i^{(t)})$, and the next variance by $\Sigma_i^{(t+1)} = h^c(\mu_i^{(t)}, \Sigma_i^{(t)})$. This gives:

$$\Delta_{\text{Dy}_2}(Z_i^{(t+1)}, h(Z_i^{(t)})) \overset{\text{def}}{=} D_{KL}(Z_i^{(t+1)}||\mathcal{N}(h^m(\mu_i^{(t)}, \Sigma_i^{(t)}), h^c(\mu_i^{(t)}, \Sigma_i^{(t)}))) \tag{7}$$

Note hat in the second case, we also make the hypothesis that the resulting distribution (for $Z_i^{(t+1)}$) is Gaussian. In the two cases, the KL divergence between the two Gaussian distributions has a simple analytic form from which the gradient can be easily computed[2].

### 3.3.3 STRUCTURAL REGULARIZATION TERM

At last, $\Delta_{\text{R}}$ corresponds to a **structural regularization** over the graph structure that encourages the model to learn similar representations for time series that are interdependent. This forces the model to learn representations that reflect the structure dependencies between the series. Recall that these

---

[2] $D_{KL}(Z_i^{(t)}||Z_j^{(t)}) = \frac{1}{2}(\text{tr}(\Sigma_j^{(t)-1}\Sigma_i^{(t)}) + (\mu_j^{(t)} - \mu_i^{(t)})^T \Sigma_j^{(t)-1}(\mu_j^{(t)} - \mu_i^{(t)}) - d - \log(\frac{|\Sigma_i^{(t)}|}{|\Sigma_j^{(t)}|}))$

dependencies are supposed to be provided as priors for this model. We define this regularization loss as:

$$\Delta_{\text{R}}(Z_i^{(t)}||Z_j^{(t)}) = D_{KL}(Z_i^{(t)}||Z_j^{(t)}) \tag{8}$$

which again has, for Gaussian random variables, a simple analytical form that can be used for learning.

Minimizing the regularization term $\Delta_{\text{R}}$ has a direct impact on the distributions of the predicted observations for connected times series. More precisely, we have the following inequality:

$$d_{TV}\left(f(Z_i^{(t)}), f(Z_j^{(t)})\right) \leqslant \sqrt{\frac{d \cdot D_{KL}(Z_i^{(t)}||Z_j^{(t)})}{2}} \tag{9}$$

with $d_{TV}$ being the total variation distance of probability measures, i.e.:

$$d_{TV}\left(X, Y\right) = \sup_{A \in \text{Borel}} \left(|\mathcal{D}_X(A) - \mathcal{D}_Y(A)|\right) \tag{10}$$

with $X$ and $Y$ being to random variables of density distribution respectively $\mathcal{D}_X$ and $\mathcal{D}_Y$, and $Borel$ being the Borel set of $\mathbb{R}^n$ (roughly, cuboids in $\mathbb{R}^n$). This means that having relatively similar representations (regarding the KL-divergence) constrains the predicted values to be similar. For more details see Appendix A.

## 3.4 INFERENCE

During inference when forecasting values, the latent distributions at $(T + 1)$ are deduced from the ones at time $T$ and follow $\mathcal{N}(h(\mu_i^{(T)}, \Sigma_i^{(T)}))$, distributions at $(T + 2)$ follow $\mathcal{N}(h \circ h(\mu_i^{(T)}, \Sigma_i^{(T)}))$, and so on...

## 4 EXPERIMENTS

### 4.1 DATASETS AND BASELINES

Experiments have been performed on four datasets respectively extracted from Google Flu Trends[3], WHO[4] and from two datasets from Grand Lyon[5] (GL) (respectively data from traffic conditions and from car parks occupancy). All the series are normalized. For all datasets, we used binary dependency relations indicating whether two series are related or not. The **Google Flu Trend** (GFT) dataset is composed of an aggregation of weekly Google search queries related to the flu in 29 countries. This dataset spans about ten years of time. The binary relations between series are defined a priori so that the series of two countries $i$ and $j$ are linked, i.e. $e_{i,j} = 1$ in Equation (1), only if the countries have a common frontier. There are 96 relations in all. The **GL Traffic** (GL-T) dataset corresponds to the traffic conditions of the 50 busiest roads of the city of Lyon (France). Data is aggregated on 20 minutes windows spanning 15 days. The binary relations between series are based on the geographical proximity of roads. There are 130 relations in total. The **GL Park** (GL-P) dataset represents the occupancy of public car parks in Lyon. The series correspond to the occupancy of the 30 busiest car parks. It has the same window and period of time as the previous dataset, and the binary relations between series are based on the geographical proximity of car parks. There are 74 relations in total. The **WHO** dataset provides the number of deaths caused by diphtheria over 91 different countries, giving rise to 91 time series. The binary relations between series are defined so that two series are linked if the corresponding countries share a common frontier. There are 228 links in total.

We compare our approach with five baselines : Auto-Regressive (**AR**), a monovariate linear auto-regressive model. It computes its predictions based on a learned linear function of a fixed number $p$ of past values of the series. The order $p$ of the model is a hyperparameter of the model selected by a grid search. Feed Forward Neural Network (**FFNN**), representative of non-linear

---

[3]http://www.google.org/flutrends
[4]http://www.who.int
[5]http://data.grandlyon.com

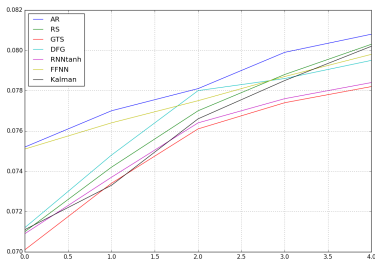

| Model | GL-T | GL-P | GFT | WHO |
|-------|------|------|-----|-----|
| AR | 0.0752 | 0.0892 | 0.0626 | 0.0832 |
| FFNN | 0.0751 | 0.0894 | 0.045 | 0.0838 |
| RNN | 0.0709 | 0.0890 | 0.0431 | **0.0795** |
| KF | 0.0711 | 0.0833 | **0.0388** | 0.0799 |
| DFG | 0.0712 | 0.0911 | 0.0592 | **0.0795** |
| $\mathbf{RDG}_{1,1}$ | 0.0742 | 0.0902 | 0.0607 | 0.0848 |
| $\mathbf{RDG}_{1,2}$ | **0.0707** | 0.0834 | 0.0434 | 0.0796 |
| $\mathbf{RDG}_{2,1}$ | 0.0765 | 0.0896 | 0.0589 | 0.0831 |
| $\mathbf{RDG}_{2,2}$ | 0.0718 | **0.0828** | 0.0429 | **0.0795** |

(a) RMSE from T+1 to T+5 on GL-T.                 (b) RMSE at T+1 on the four datasets.

Figure 1: Quantitative comparison between baselines and our proposed model (RDG) on the prediction task. $\mathrm{RDG}_{k,l}$ corresponds to the variant with losses ($\Delta_{\mathrm{De}_k}$, $\Delta_{\mathrm{Dy}_l}$).

auto-regressive models of order $p$ where the non-linear function is modeled as a feed-forward neural network with one hidden layer of size $s$. In this case, $p$ and $s$ are hyperparameters selected by grid search. **RNN**, a recurrent neural network with one hidden layer of size $s$ of recurrent units and tanh non-linearities. The RNN model is a state space non-linear auto-regressive model with exogenous inputs (the past values of the series). Note that this model should in principle be able to learn the inter-series dependencies, but the dependencies are not modeled explicitly as they are in our model. Also the RNN does not introduce explicit modeling of uncertainties. **KF** (Kalman (1960)), is a classic Kalman Filter with linear transformations from one state to another. **DFG** (Mirowski & LeCun (2009)), a state of the art model that learns continuous deterministic latent variables by modeling the dynamics and the joint probabilities between series. All the hyperparameters of the baselines have been set using a validation set by grid search, including the best architectures for the dynamic model $h$ when it is a multi-layer perceptron with one hidden layer or a linear model. .

For the evaluation we have considered a root-mean-square error (RMSE) criterion. Regarding the experimental protocol, models are evaluated using cross-validation with rolling origin.

## 4.2 RESULTS

Let us first present the performance of our model w.r.t. the baselines for prediction at horizon 1 in Figure 1b We have tested the four variants of our approach i.e combinations of $\Delta_{\mathrm{De}_1}$ or $\Delta_{\mathrm{De}_2}$ with $\Delta_{\mathrm{Dy}_1}$ or $\Delta_{\mathrm{Dy}_2}$. The proposed model obtains the best results on all the datasets except GFT where KF performs better. Otherwise it outperforms the baselines on two datasets (GL-P -Grand Lyon Parks- and GFT -Google Flu Trends- on the table) and gets results similar to the RNN on the two others (GL-T -Grand yon Traffic- and WHO). The non linear dynamical model used for $\Delta_{\mathrm{Dy}_2}$ usually gets better results than other models, the best combination being the use of the MSE expectation error for the decoder and the non-linear model for the dynamics (denoted $\mathbf{RDG}_{2,2}$ on the figure). The non linear dynamical model used for $\Delta_{\mathrm{Dy}_2}$ usually gets better results than other models, the best combination being the use of the MSE expectation error for the decoder and the non-linear model for the dynamics (denoted $\mathbf{RDG}_{2,2}$ on the figure).

Figure 1a shows the prediction quality (RMSE) at $(T+1)$, $(T+2)$, $(T+3)$, $(T+4)$ and $(T+5)$ and illustrates the ability of RDG to predict correctly at different horizons. Here again, the performance of RDG is very close to the performance of the Recurrent Neural Network. One can remark that at $(T+5)$ KF does not goes the distance since it performs well at $(T+1)$ but quite badly at $(T+5)$ in comparison to other baselines.

RDG has the additional property of modeling the uncertainty associated to its predictions, which is not the case for a RNN. Let us consider the curves presented in Figure 2. They illustrate, the predictions made by our model together with their associated variance computed through the Gaussian embeddings. First, one can see that the ground truth values are always within the confidence interval provided by our model, which means that RDG computes relevant minimum and maximum possible values. Another aspect is that the size of the interval increases with the prediction horizon, which is

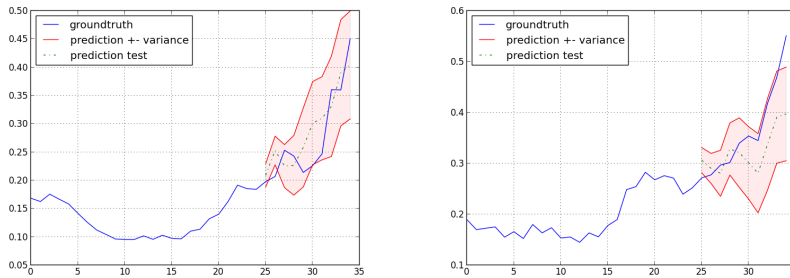

Figure 2: Forecasts on GFT (two different time series of the dataset) with the $\text{RDG}_{2,2}$ model showing its range of confidence: $E(f(Z^{(t)})) \pm \text{var}(f(Z^{(t)}))$. Prediction at $25+n$ corresponds to $f(h^n(Z^{(25)}))$.

what is expected from such a model. The latter is then able to predict relevant confidence values for its predictions.

**Comparison between RDG with/without structural regularization or uncertainty**   We compare in Table 1 the results between our model when taking into account the neighborhood graph ($\lambda_R \neq 0$) or not ($\lambda_R = 0$): forecasts are uniformly worse for all datasets when we do not take into account the neighborhood graph, it suggests that the regularizer improves the model when the input graph is relevant. Furthermore, we give the results obtained without uncertainty, which corresponds to the model described in (Ziat et al. (2016)) (denoted Rainstorm): here again, our model outperforms the previous one for all the datasets.

| Model | Dataset | | | |
|---|---|---|---|---|
| | GL-T | GL-P | GFT | WHO |
| Rainstorm | 0.0710 | 0.0886 | 0.0440 | 0.0804 |
| RDG ($\lambda_R = 0$) | 0.0719 | 0.900 | 0.0441 | 0.0807 |
| RDG | **0.0707** | **0.0828** | **0.0388** | **0.0795** |

Table 1: RMSE at $T+1$ on the four datasets

## 5   CONCLUSION AND FUTURE WORK

We have proposed a model for relational time series forecasting. Our model (RDG) is based on latent Gaussian embeddings, and has shown competitive performance on four different datasets compared to state-of-the-art models. Moreover, RDG allows us to model the uncertainty of predictions, providing for example confidence intervals for each prediction. Future work will investigate more complex dynamic and prediction functions, as well as observing the behavior of the model for imputation tasks.

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

## A  IMPACT OF MINIMIZING THE KL-DIVERGENCE ON PREDICTED VALUES

In this section, we show that the structural regularization term between two time series bounds the difference predicted observations.

Since we use diagonal covariance matrices and that the KL-divergence is invariant by multiplying both random variables by the same scalar, we can show that:

$$D_{KL}(Z_i^{(t)}||Z_j^{(t)}) = \sum_{k=1}^{d} D_{KL}(Z_{i,k}^{(t)}||Z_{j,k}^{(t)}) = \sum_{k=1}^{d} D_{KL}(\theta_k Z_{i,k}^{(t)}||\theta_k Z_{j,k}^{(t)}) \tag{11}$$

with $Z_{i,k}^{(t)}$ being the $k$-th component of the gaussian vector $Z_i^{(t)}$.

Then, using Pinsker's inequality one can see that minimizing the KL-divergence also minimize the total variation norm (which can be more intuitive in some cases), leading to:

$$2\sum_{k=1}^{d} \left( d_{TV}(\theta_k Z_{i,k}^{(t)}, \theta_k Z_{j,k}^{(t)}) \right)^2 \leqslant \sum_{k=1}^{d} D_{KL}(\theta_k Z_{i,k}^{(t)}||\theta_k Z_{j,k}^{(t)}) \tag{12}$$

with $d_{TV}$ being the total variation distance of probability measures.

Using the Cauchy–Schwarz inequality:

$$\frac{1}{d} \left( \sum_{k=1}^{d} d_{TV}(\theta_k Z_{i,k}^{(t)}, \theta_k Z_{j,k}^{(t)}) \right)^2 \leqslant \sum_{k=1}^{d} \left( d_{TV}(\theta_k Z_{i,k}^{(t)}, \theta_k Z_{j,k}^{(t)}) \right)^2 \tag{13}$$

Finally, each component of the random vectors $Z^{(t)}$ being pairwise independent, we have:

$$d_{TV}(\sum_{k=1}^{d} \theta_k Z_{i,k}^{(t)}, \sum_{k=1}^{d} \theta_k Z_{j,k}^{(t)}) \leqslant \sum_{k=1}^{d} d_{TV}(\theta_k Z_{i,k}^{(t)}, \theta_k Z_{j,k}^{(t)}) \tag{14}$$

Combining the the inequalities above, we can straightforwardly show the following inequality:

$$d_{TV}\left( f(Z_i^{(t)}), f(Z_j^{(t)}) \right) \leqslant \sqrt{\frac{d \cdot D_{KL}(Z_i^{(t)}||Z_j^{(t)})}{2}} \tag{15}$$

