# Peer review of "Modelling Relational Time Series using Gaussian Embeddings"

_ICLR 2017 — rejected_

[Official Review · AnonReviewer3 · rating 4 · confidence 5 · 16 Dec 2016 (modified: 20 Jan 2017)]
**Interesting model, further experiments required**

In absence of authors' response, the rating is maintained.

---

This paper introduces a nonlinear dynamical model for multiple related multivariate time series. It models a linear observation model conditioned on the latent variables, a linear or nonlinear dynamical model between consecutive latent variables and a similarity constraint between any two time series (provided as prior data and non-learnable). The predictions/constraints given by the three components of the model are Gaussian, because the model predicts both the mean and the variance or covariance matrix. Inference is forward only.

The model is evaluated on four datasets, and compared to several baselines: plain auto-regressive models, feed-forward networks, RNN and dynamic factor graphs DFGs, which are RNNs with forward and backward inference of the latent variables.

The model, which introduces lateral constraints between different time series, and which predicts both the mean and covariance seems interesting, but presents two limitations.

First of all, the paper should refer to variational auto-encoders / deep gaussian models, which also predict the mean and the variance during inference.

Secondly, the datasets are extremely small. For example, the WHO contains only 91 times series of 52*10 = 520 time points. Although the experiments seem to suggest that the proposed model tends to outperform RNNs, the datasets are very small and the high variance in the results indicates that further experiments, with longer time series, are required. The paper could also easily be extended with more information about the model (what is the architecture of the MLP) as well as time complexity comparison between the models (especially between DFGs and this model).

Minor remark:
The footnote 2 on page 5 seems to refer to the structural regularization term, not to the dynamical term.

[Official Review · AnonReviewer2 · rating 4 · confidence 3 · 17 Dec 2016 (modified: 21 Jan 2017)]
**Important line of research, muddled presentation and unconvincing empirical results**

Because the authors did not respond to reviewer feedback, I am maintaining my original review score.

-----

This paper proposes to model relational (i.e., correlated) time series using a deep learning-inspired latent variable approach: they design a flexible parametric (but not generative) model with Gaussian latent factors and fit it using a rich training objective including terms for reconstruction (of observed time series) error, smoothness in the latent state space (via a KL divergence term encouraging neighbor states to be similarly distributed), and a final regularizer that encourages related time series to have similar latent state trajectories. Relations between trajectories are hard coded based on pre-existing knowledge, i.e., latent state trajectories for neighboring (wind speed) base stations should be similar. The model appears to be fit using gradient simple descent. The authors propose several elaborations, including a nonlinear transition function (based on an MLP) and a reconstruction error term that takes variance into account. However, the model is restricted to using a linear decoder. Experimental results are positive but not convincing.

Strengths:
- The authors target a worthwhile and challenging problem: incorporating the modeling of uncertainty over hidden states with the power of flexible neural net-like models.
- The idea of representing relationships between hidden states using KL divergence between their (distributions over) corresponding hidden states is clever. Combined with the Gaussian distribution over hidden states, the resulting regularization term is simple and differentiable.
- This general approach -- focusing on writing down the problem as a neural network-like loss function -- seems robust and flexible and could be combined with other approaches, including variants of variational autoencoders.

Weaknesses:
- The presentation is a muddled, especially the model definition in Sec. 3.3. The authors introduce four variants of their model with different combinations of decoder (with and without variance term) and linear vs. MLP transition function. It appears that the 2,2 variant is generally better but not on all metrics and often by small margins. This makes drawing a solid conclusions difficult: what each component of the loss contributes, whether and how the nonlinear transition function helps and how much, how in practice the model should be applied, etc. I would suggest two improvements to the manuscript: (1) focus on the main 2,2 variant in Sec. 3.3 (with the hypothesis that it should perform best) and make the simpler variants additional "baselines" described in a paragraph in Sec. 4.1; (2) perform more thorough experiments with larger data sets to make a stronger case for the superiority of this approach.
- The authors only allude to learning (with references to gradient descent and ADAM during model description) in this framework. Inference gets its one subsection but only one sentence that ends in an ellipsis (?).
- It's unclear what is the purpose of introducing the inequality in Eq. 9.
- Experimental results are not convincing: given the size of the data, the differences vs. the RNN and KF baselines is probably not significant, and these aren't particularly strong baselines (especially if it is in fact an RNN and not an LSTM or GRU).
- The position of this paper is unclear with respect to variational autoencoders and related models. Recurrent variants of VAEs (e.g., Krishnan, et al., 2015) seem to achieve most of the same goals as far as uncertainty modeling is concerned. It seems like those could easily be extended to model relationships between time series using the simple regularization strategy used here. Same goes for Johnson, et al., 2016 (mentioned in separate question).

This is a valuable research direction with some intriguing ideas and interesting preliminary results. I would suggest that the authors restructure this manuscript a bit, striving for clarity of model description similar to the papers cited above and providing greater detail about learning and inference. They also need to perform more thorough experiments and present results that tell a clear story about the strengths and weaknesses of this approach.

[Official Review · AnonReviewer1 · rating 4 · confidence 4 · 17 Dec 2016]
**Interesting idea but formulation and experiments not convincing**

This manuscript proposes an approach for modeling correlated timeseries through a combination of loss functions which depend on neural networks. The loss functions correspond to: data fit term, autoregressive latent state term, and a term which captures relations between pairs of timeseries (relations have to be given as prior information).

Modeling relational timeseries is a well-researched problem, however little attention has been given to it in the neural network community. Perhaps the reason for this is the importance of having uncertainty in the representation. The authors correctly identify this need and consider an approach which considers distributions in the state space.

The formulation is quite straightforward by combining loss functions. The model adds to Ziat et al. 2016 in certain aspects which are well motivated, but unfortunately implemented in an unconvincing way. To start with, uncertainty is not treated in a very principled way, since the inference in the model is rather naive; I'd expect employing a VAE framework [1] for better uncertainty handling. Furthermore, the Gaussian co-variance collapses into a variance, which is the opposite of what one would want for modelling correlated time-series. There are approaches which take these correlations into account in the states, e.g. [2].

Moreover, the treatment of uncertainty only allows for linear decoding function f. This significantly reduces the power of the model. State of the art methods in timeseries modeling have moved beyond this constraint, especially in the Gaussian process community e.g. [2,3,4,5]. Comparing to a few of these methods, or at least discussing them would be useful.


References:
[1] Kingma and Welling. Auto-encoding Variational Bayes. arXiv:1312.6114
[2] Damianou et al. Variational Gaussian process dynamical systems. NIPS 2011.
[3] Mattos et al. Recurrent Gaussian processes. ICLR 2016.
[4] Frigola. Bayesian Time Series Learning with Gaussian Processes, University of Cambridge, PhD Thesis, 2015. 
[5] Frigola et al. Variational Gaussian Process State-Space Models. NIPS 2014


One innovation is that the prior structure of the correlation needs to be given. This is a potentially useful and also original structural component. However, it also constitutes a limitation in some sense, since it is unrealistic in many scenarios to have this prior information. Moreover, the particular regularizer that makes "similar" timeseries to have closeness in the state space seems problematic. Some timeseries groups might be more "similar" than others, and also the similarity might be of different nature across groups. These variations cannot be well captured/distilled by a simple indicator variable e_ij. Furthermore, these variables are in practice taken to be binary (by looking at the experiments), which would make it even harder to model rich correlations. 

The experiments show that the proposed method works, but they are not entirely convincing. Importantly, they do not shed enough light into the different properties of the model w.r.t its different parts. For example, the effect and sensitivity of the different regularizers. The authors state in a pre-review answer that they amended with some more results, but I can't see a revision in openreview (please let me know if I've missed it). From the performance point of view, the results are not particularly exciting, especially given the fact that it's not clear which loss is better (making it difficult to use the method in practice). 

It would also be very interesting to report the optimized values of the parameters \lambda, to get an idea of how the different losses behave.

Timeseries analysis is a very well-researched area. Given the above, it's not clear to me why one would prefer to use this model over other approaches. Methodology wise, there are no novel components that offer a proven advantage with respect to past methods. The uncertainty in the states and the correlation of the time-series are the aspects which could add an advantage, but are not adequately researched in this paper.

[Final Decision · Program Chairs · 06 Feb 2017]
**ICLR committee final decision**

The reviews of this paper seem to be very aligned: many of the ideas presented in the paper are interesting, the problem is important, and the results encouraging but preliminary. R2 thought the paper could be improved in terms of clarity and offered several specific suggestions to this end. R2 and R1 mentioned the limitations of the linear decoder; which is not a critical flaw, in my opinion, but as R1 points out, many recent works have explored nonlinear decoders and these could be at least discussed, if not compared. All of the reviewers have worked in this area and expressed high-confidence reviews.
 
 I was surprised that the authors did not provide feedback or revise the paper at least with reference to the clarity/presentation suggestions. It seems this may have had an impact on the perception of the reviewers. I encourage the authors to revise the paper in light of the reviews and re-submit to another venue.